# Study on Mechanical Properties of Nanopores in CoCrFeMnNi High-Entropy Alloy Used as Drug-Eluting Stent

**DOI:** 10.3390/ma17133314

**Published:** 2024-07-04

**Authors:** Zhen Zhou, Chaoyue Ji, Dongyang Hou, Shunyong Jiang, Zihan Yang, Fang Dong, Sheng Liu

**Affiliations:** 1The Institute of Technological Sciences, Wuhan University, Wuhan 430072, China; zhen.z@whu.edu.cn (Z.Z.); 2018106520021@whu.edu.cn (C.J.); 2022106520011@whu.edu.cn (D.H.); 2023206520002@whu.edu.cn (S.J.); 2023206520022@whu.edu.cn (Z.Y.); 2Wuhan Institute of Quantum Technology, Wuhan 430206, China; 3School of Power and Mechanical Engineering, Wuhan University, Wuhan 430072, China

**Keywords:** mechanical properties, high-entropy alloy, drug-eluting stent, molecular dynamics, nanopores

## Abstract

The CoCrFeMnNi high-entropy alloy is commonly used for vascular stents due to its excellent mechanical support and ductility. However, as high-entropy alloy stents can cause inflammation in the blood vessels, leading to their re-narrowing, drug-eluting stents have been developed. These stents have nanopores on their surfaces that can carry drug particles to inhibit inflammation and effectively prevent re-narrowing of the blood vessels. To optimize the mechanical properties and drug-carrying capacity of high-entropy alloy stents, a high-entropy alloy system with different wide and deep square-shaped nanopore distributions is created using molecular dynamics. The mechanical characteristics and dislocation evolution mechanism of different nanopore high-entropy alloy systems under tensile stress were studied. The results showed that the CoCrFeMnNi high-entropy alloy with a rational nanopore distribution can effectively maintain the mechanical support required for a vascular stent. This research provides a new direction for the manufacturing process of nanopores on the surfaces of high-entropy alloy stents.

## 1. Introduction

The global prevalence of cardiovascular disease has increased in the past three decades, increasing from 271 million cases in 1990 to 523 million cases in 2019, with 18.9 million deaths, or one-third of the total deaths [1,2]. Currently, the most effective treatment for vascular thrombosis disease is interventional therapy, such as installing vascular bare-metal stents. Compared with traditional stainless steel vascular bare-metal stents, CoCrFeMnNi high-entropy alloy vascular bare-metal stents are widely used in the biomedical field, especially in vascular interventional therapy, due to their more robust mechanical support performance and good biological compatibility and impenetrability to rays [3].

However, the biggest problem to be overcome is that the implantation of the stent leads to restenosis of the vascular lesion. With the progressive development of vascular interventional therapy, drug-elution stents are also widely used. Drug-elution stents can increase the amount of drug (sirolimus) loading while slowing down the drug release rate and potentially eliminating the polymer, all of which are expected to aid in preventing late stent thrombosis [4]. According to statistics, the stenosis rate of vascular lesions in bare metal stents is 20–30% [5]. At the same time, the stenosis rate in drug stents is 10–18% lower than that in bare stents [6]. A drug-eluting stent acts on a lesion vascular stent owing to the performance of the vascular stent on the surface. At the same time, the surface of the drug-eluting stent consists of anti-thrombosis drugs to prevent inflammatory complications and coagulation at the position of vascular and stent action [7,8]. Therefore, the drug-eluting stents are widely used. Commonly used drug-elution stents should have excellent drug-carrying ability and good support properties. Thus, while satisfying the ability of vascular stents to carry drugs, stents remain crucial for improving the mechanical properties of the narrow vascular support.

Recent studies have focused on drug-eluting stents’ ability to evaluate lesion blood vessels. The main focus has been on biological research, such as studying a drug stent for elution [9,10] and different components of drugs to regulate and resist the inflammatory response inside blood vessels [11,12].

Moreover, a porous titanium dioxide coating has been coated on a 316L stainless steel stent using magnetron sputtering and plasma oxidation technology to obtain a porous drug-loaded vascular stent [13]. In addition, the width and surface roughness of a vascular stent were investigated by adjusting the pulse laser power [14].

Nevertheless, most studies have focused on the mechanical studies of bare stents, such as the mechanical structure and properties of the stents [15,16,17]. Only a few studies have focused on the structural mechanics of drug-eluting stents, especially the influence of nanopore distribution on the mechanical properties of vascular stents. Others only analyze the evolution of the single voids and elucidate their effects on the deformation mechanism of the single-crystal CoCrFeMnNi high-entropy alloy under tension [18,19]. Due to the accidental nature of the study of crystal defects by a single hole, these studies often ignore the concentrated influence of multiple holes on the mechanical properties of high-entropy alloys.

To better understand the effect of multiple surface nanopores on the mechanical properties of vascular drug-loaded stents, this paper analyzes the mechanics and microstructure evolution of nanoporous drug-loaded scaffolds in the process of expansion from the perspective of molecular dynamics using high-entropy alloys with square group pores of different depths and widths. The high-entropy alloy CoCrFeMnNi was selected as the research material because material L605 uses the high-entropy alloy composition. The high-entropy alloy L605 is compatible and supports smooth muscle cells triggered by vascular injury [20]. Compared to stainless-steel stents, high-entropy alloy L605 stents have been widely used because of their excellent mechanical and radiopaque properties and good biocompatibility [20]. The nanopores in vascular stents expand along with the stent, and their shape changes with the size and mechanical properties. The change in the shape of the nanopores affects the release of the loaded particles. As shown in Figure 1, the CoCrFeMnNi high-entropy alloy vascular stent containing nanopores carrying drug particles expands the vessel in the stenosis, and the nanopores on the vascular stent change accordingly. The CoCrFeMnNi high-entropy alloy vascular stent containing nanopores carrying drug particles expanded the blood vessels at the narrowing point, which corresponded to changes in the nanopores in the vascular stent when the CoCrFeMnNi high-entropy alloy was stretched in both directions on the atomic scale.

In this study, we performed an MD simulation to study the influence law of different square-shaped pore size widths and depths of nanopores in the lower high-entropy alloy on the mechanical mechanism of the CoCrFeMnNi high-entropy alloy and the evolution mechanism of dislocations during the stretching process [21].

## 2. Modeling and Simulation Methods

### 2.1. Force Potential

To study the mechanical properties of CoCrFeMnNi high-entropy alloy nanopores at the atomic scale, we adopted the potential function of the interaction between atoms in the alloy according to reference [22]. The pure component crystal structure for studying the long-range Lennard–Jones potential was characterized using the potential function of the CoCrFeMnNi high-entropy alloy [22]. The interaction energy model of the atoms is based on the lj truncation potential function model, where all the interaction energies are between a single atom and a neighboring atom, and it can be expressed using the following Lennard–Jones model equation:(1)$$E_{at}=2\varepsilon \sum_{i}\left[{\left(\frac{\sigma }{r_{i}}\right)}^{12}-{\left(\frac{\sigma }{r_{i}}\right)}^{6}\right].$$

The equation represents the sum of interaction energies of all nearest-neighbor atoms centered on or adjacent to *i*. In Equation (1), *r_i_* represents the cut-off radius between a specific nearest neighbor atom and an *i* atom in the crystal structure. σ and ε represent two related parameters in the Lennard–Jones potential function expression. Through nearest-neighbor correlation coefficient conversion [22,23], we could obtain the following:(2)$$E_{at}=2\varepsilon \left[\sum_{i}\left[\frac{1}{a_{i}^{12}}\right]\left(\frac{\sigma }{r_{0}}\right)-\sum_{i}\left[\frac{1}{a_{i}^{6}}\right]{\left(\frac{\sigma }{r_{0}}\right)}^{6}\right].$$
where *r*_0_ is the first nearest distance from the central atom *i*, *a_i_* is the ratio coefficient between *r*_0_ and *r_i_*, and the relationship between *a_i_* and *r_i_* is *r_i_* = *a_i_r*_0_. By setting $A_{12}=\underset{i}{\sum }\left[\frac{1}{a_{i}^{12}}\right]$ and $A_{6}=\underset{i}{\sum }\left[\frac{1}{a_{i}^{6}}\right]$, Equation (2) can be converted to the Lennard–Jones potential function equation related to *E_at_* and parameter *ε* [23,24]:(3)$$\varepsilon =-\frac{2A_{12}}{A_{6}^{2}}E_{at}.$$

When the equilibrium was in a single atom pure component system, $\frac{dE_{at}}{dr_{0}}\;$= 0. At this point, the closest distance between atom i and the adjacent atom was $r_{0}$. By introducing the Lennard–Jones parameter *σ* in a single atom pure component system, the following can be found:(4)$$r_{0}=\sigma {\left(\frac{2A_{12}}{A_{6}}\right)}^{\frac{1}{6}}.$$

For a single atom pure component system, there are equations for density calculations and spherical volume calculations $r_{0}={\left(\frac{mN_{cell}}{\beta \rho }\right)}^{\frac{1}{3}}$, where m is the total mass of the pure component system of a single atom, and *β* is a crystal structure factor in a single atom pure component system. *ρ* is the density of the pure component system of a single atom. *N_cell_* is the total number of atoms in a single atom pure component system, where the cell volume expression is $V_{cell}=\beta r_{0}^{3}$. The potential function of the CoCrFeMnNi high-entropy alloy was treated directly as a regular solution [24,25,26]. Finally, by introducing the Lorentz–Berthelot hybrid rule, the relevant parameters and the truncation radius of adjacent atoms in the potential function of the alloy were calculated in Table 1 [22].

### 2.2. Method and Software

In this study, the relaxation process of the CoCrFeMnNi high-entropy alloy was 300 K, and the total relaxation time was 40 ps [18]. The energy minimization was performed using the conjugate gradient algorithm [27]. To better characterize the mechanical parameters of the material, we used quasi-static stretching. The direction of biaxial stretching was in the *y*-axis direction, and the flow of simulating stretching is shown in Figure 2.

The dimension of the simulation model was 18 × 18 × 18 nm as the unit cell system [28]. CoCrFeMnNi is a high-entropy alloy with a medical stent composition [20]. The contents of each element in the CoCrFeMnNi high-entropy alloy are shown in Table 2.

In this study, the control variable method was used to examine the effects of the nanopores on the mechanical properties of the CoCrFeMnNi high-entropy alloy. The main control factors for the nanopores included the size, depth, and strain rate. The different factors are shown in Table 3.

Using Table 3, we show the specific classifications in Table 4. To study the effect of different strain rates on the mechanical properties of the high-entropy alloy containing nanopores, we used ①, ②, ③, ④, ⑤, and ⑥ to study the effect of different strain rates on the mechanical properties of the alloy. Then, ⑦, ⑧, ⑨, and ⑩ were used to study the influence of the width of different apertures on the mechanical properties of the alloys containing nanopores. Finally, ⑪, ⑫, ⑬, and ⑭ were used to study the effect of varying hole depths on the mechanical properties of the high-entropy alloys. Molecular dynamics simulations were performed using Lammps Stable Release [21], and the results were visualized using Ovito 3.0.0 [29].

## 3. Results and Discussion

Because the tensile force on the crystal cell structure of the CoCrFeMnNi high-entropy alloy with nanopores was along the *y*-axis direction, we can conclude from the si*x*-axis stress–strain curve (shown in Figure 3) from the stretching results that the stress on the crystal cell structure mainly originates from the y–y direction, and the stress in other directions fluctuates around 0 GPa. Therefore, the stress direction of the mechanical performance studied in this article was primarily along the y–y direction.

### 3.1. Analysis of the Stress Evolution Mechanism of the High-Entropy Alloy Containing Nanopores under Different Tensile Strain Rates

The generated strain rate considerably affects the mechanical properties of nanopore CoCrFeMnNi high-entropy alloy stent from the release and expansion of containing at the site of vascular lesions. Therefore, analyzing the effect of strain rate on the mechanical properties of the CoCrFeMnNi high-entropy alloy containing nanopores is essential to understanding the release of a loaded drug from an alloy stent at the site of vascular stenosis. The stress–strain curves of the CoCrFeMnNi high-entropy alloy containing nanopores under different loading strain rates were analyzed, as shown in Figure 4.

Figure 4 shows that the peak stress and strain of the CoCrFeMnNi high-entropy alloy containing nanopores gradually increased with the strain rate below 0.02 ps^−1^. When the strain rate was 0.02 ps^−1^, the peak stress reached a maximum of 7.186 GPa. However, when the strain rate exceeded 0.02 ps^−1^, the peak stress decreased with an increasing strain rate. However, the strain increased as the strain rate increased. The CoCrFeMnNi high-entropy alloy containing nanopores exhibited considerable toughness characteristics overall. To further investigate the deformation mechanism caused by different loading strain rates, we obtained the average stress cloud map of the CoCrFeMnNi high-entropy alloy unit cell system containing nanopores during the stretching process. We analyzed the evolution mechanism of the average principal stress of the high-entropy alloy cell system containing nanopores during the stretching process. Because the influence of shear stress on the principal stress was relatively small, according to the si*x*-axis stress–strain curve in Figure 3, we focused on the evolution mechanism of the average stress during the tensile process by studying the principal stress.

The equation for calculating the atomic average principal stress of the CoCrFeMnNi high-entropy alloy with nanopores is as follows:(5)$$\bar{\sigma }=\frac{\sigma_{xx}+\sigma_{xx}+\sigma_{xx}}{30000V}.$$
where $\bar{\sigma }$ represents the mean principal stress, and $\sigma_{xx}$, $\sigma_{yy}$, and $\sigma_{zz}$ represent the primary stress in the *x*, *y*, and *z* directions, respectively. V indicates the volume of the CoCrFeMnNi high-entropy alloy cells containing nanopores during the tensile process. The real-time tensile average principal stress cloud map was obtained at different strain rates using Equation (5). Figure 4 shows the average principal stress contour of tensile peak stress before and after the tensile peak stress points.

As shown in Figure 5, the average principal stress cloud map at different strain rates ranged from 0.001 to 0.05 ps^−1^, and the strain corresponding to peak stress increased gradually from 0.055 to 0.125. Meanwhile, the strain corresponding to peak stress increased gradually. When the strain rate gradually increased from 0.001 to 0.02 ps^−1^, the corresponding peak stress rose from 5.20 to 7.18 GPa. Thus, the peak stress increased as the strain rate increased. When the strain rate gradually increased from 0.02 to 0.05 ps^−1^, the corresponding peak stress decreased from 7.18 to 4.49 GPa. Meanwhile, the peak stress decreased as the strain rate increased, and the CoCrFeMnNi high-entropy alloy unit cell system containing nanopores exhibited strong toughness characteristics, which was more susceptible to deformation. With a significant strain rate, the high-entropy alloy unit cell system containing nanopores was more rigid and more prone to deformation. During the stretching process, the average principal stress cloud map showed that the average principal stress before the peak stress point to the peak stress point gradually increased and that the average principal stress was focused on the Y direction of the stretch. Moreover, according to the average principal stress cloud map, the average principal stress was concentrated at the edge of the unit cell system along the tensile direction and at the edge of the nanopores.

### 3.2. Tensile Mechanics Evolution of the CoCrFeMnNi High-Entropy Alloy with Different Depths of Nanopores

Nanopores of different depths were studied to analyze further the evolution mechanism of the nanopore edge stress concentration and the mechanical variation of the nanopores in the tensile process. The analysis in Figure 5 shows that, when the strain rate is 0.05 ps^−1^, a significant strain rate, the high-entropy alloy unit cell system containing nanopores was more rigid and more prone to deformation. The constant strain rate was 0.05 ps^−1^, and the aperture width was 20 nm. The CoCrFeMnNi high-entropy alloy unit cell system was stretched with depths of 10, 20, 30, and 40 nm nanopores. We obtained the stress–strain curve along the tensile direction by analyzing the principal stresses in the Y direction, as shown in Figure 6.

In Figure 6, when the aperture depths of 10, 20, 30, and 40 Å increased gradually, the corresponding peak stresses in the stress–strain curve were 6.69, 4.49, 4.52, and 4.58 GPa, respectively. As shown in Figure 6, stress was linearly correlated with strain when the strain was less than 0.05. The elastic behavior was evident, and the elastic modulus was significantly higher than 20 Å when the depths of the nanopore were 10, 30, and 40 Å. In addition, it can be seen in Figure 6 that when the aperture depth was 20, 30, and 40 Å, the slope performance of the curve at the elastic stage was not significantly different. The peak stress in the 30 and 40 Å stress–strain curves was also close to 4.5 GPa. We concluded that the mechanical properties of the CoCrFeMnNi high-entropy alloy containing nanopores were not sensitive to depth.

Upon studying the mechanical deformation law of the CoCrFeMnNi high-entropy alloy at different depths, we found that the strain curve analysis yielded values of 0.05, 0.1, 0.15, and 0.2, and the corresponding average principal stress cloud map before and after stretching is shown in Figure 7.

When the nanopore depth was 10 nm, the peak stress was up to 6.48 GPa. Combined with the analysis in Figure 6 of the different depth tensile stress–strain curves, the CoCrFeMnNi high-entropy alloy cell system showed strong yield properties. As the strain gradually increased from 0 to 0.15, the maximum average principal stress increased gradually, and the average principal stress was primarily concentrated in the edge of the nanopores along the tensile direction. In the overall tensile process, the shape of the nanopores did not considerably change, even at the last strain of 0.2. The CoCrFeMnNi high-entropy alloy unit cell system containing nanopores showed a state of tensile fracture. The nanopores did not show significant deformation characteristics. When the depth of the nanopores continued to increase to 20 nm at the beginning of the stretch, the average principal stress was primarily concentrated at both ends of the tensile direction. When the strain was 0.1, the average principal stress was concentrated in the middle of the nanopores of the CoCrFeMnNi high-entropy alloy. As the stretch continued, the average principal stress progressively extended along the Y-tensile direction to the edge of the entire Y-tensile direction. During the tensile process, when the strain reached 0.15, the average principal stress peaked at 3.98 GPa. The average principal stress was significantly lower at its peak than when the depth was 10 nm. When the depths were 30 and 40 nm, the average principal stress was concentrated at both ends of the tensile direction at the beginning of the stretch. When the strain was 0.1, the average principal stress was concentrated in the middle of the nanopores of the CoCrFeMnNi high-entropy alloy. As the stretch continued, the average principal stress progressively extended along the Y-tensile direction to the edge of the entire Y-tensile direction. During stretching, when the strain was the same, the average principal stress was significantly lower at depths of 20, 30, and 40 nm than at a depth of 10 nm and was considerably different. However, the average sizes of the principal stresses of the CoCrFeMnNi high-entropy alloys at depths of 20, 30, and 40 nm for the nanopores were close when the same strain was present in the tensile process. In the overall tensile process, the shape of the nanopore did not appear to change considerably. Furthermore, when the depth of the nanopore exceeded 10 nm, increasing the depth of the nanopore had little effect on the mechanical properties of the nanopores of the CoCrFeMnNi high-entropy alloys.

#### Crystal Structure and Dislocation Evolution Mechanism of the CoCrFeMnNi High-Entropy Alloy Containing Nanopores at Different Depths

To further analyze the influence of the structure and dislocation evolution mechanism on the mechanical properties of the nanopores of the CoCrFeMnNi high-entropy alloys, we performed a phase transition analysis of the alloy with nanopores at different depths at the tensile strain rate of 0.05 ps^−1^. The crystal structure percentages of the nanopores at varying depths in the CoCrFeMnNi high-entropy alloy crystals, with strains of 0.05, 0.1, 0.15, and 0.2, are shown in Figure 6.

The crystal structures of the nanopores in the CoCrFeMnNi high-entropy alloy included face center cubic (FCC), hexagonal close-packed (HCP), and other (amorphous structures). Upon the horizontal comparative analysis of Figure 8a–d, with increasing strain from 0.05 to 0.2, we found that the percentage of FCC structures was always the largest and that the percentage of HCP crystal structures was always the smallest. The nanopores in the CoCrFeMnNi high-entropy alloy had primarily an FCC structure. When the depth of the nanopores was 10 nm, the percentage of FCC, HCP, and other crystal structures did not change significantly during the tensile process. When the depths of the nanopores were 20, 30, and 40 nm, it increased gradually, and the percentage of the FCC crystal structure gradually decreased. Meanwhile, the percentage of other crystal structures increased, and the change was more pronounced with an increase in the depth of the nanopores. Moreover, the HCP percentage of the crystal structure remained almost unchanged. Upon the longitudinal comparative analysis of Figure 8a–d, because the CoCrFeMnNi high-entropy alloy primarily had an FCC crystal structure, we found that when the strain reached a specific value, as the depth of the nanopore increased, the percentage of FCC in the alloy overall gradually decreased. In contrast, the percentage of other crystal structures gradually increased. However, the percentage of the HCP crystal structure remained almost unchanged. Further analysis of the transformation evolution showed that the FCC crystal structure content decreased, and other crystal structures increased. The radial distribution function of the high-entropy alloy with different depths of nanopores was obtained, as shown in Figure 9.

The crystal structure of the nanopores in the CoCrFeMnNi high-entropy alloy at different depths under different strains was further analyzed using the RDF waterfall diagram shown in Figure 9.

In Figure 9a–d, the waterfall diagram shows that the characteristic peaks were both in the long and short ranges. Furthermore, the long and short ranges were orderly, but the peak length differed. It was a polycrystalline structure. For nanopores of the same depth, with increasing strain, the characteristic peaks did not increase or decrease during the tensile process. However, the first characteristic peak appeared at 2.525 Å, and the third characteristic peak at 4.325 Å. In particular, the first characteristic peak was more prominent. For nanopores of the same depth, the intensities of different peaks increased significantly as the strain increased. Moreover, the atoms of the characteristic peaks changed accordingly. In particular, the strain reached 0.25 at various depths. The intensities of the characteristic peaks of the Ni–Ni paired atoms considerably increased and exceeded those of the other peaks, which indicated that the movement of Ni atoms in the tensile process affected the tightness of the atomic arrangement, the ordering of the cell structure, and the dislocation slip crystal structure. During the tensile crystal structure process, atoms changed accordingly. To analyze this evolution law further, we analyzed the dislocation of nanopores under different strains at different depths, as shown in Figure 10.

Figure 10 shows the tensile edge of the nanopores of the CoCrFeMnNi high-entropy alloy. The Shockley dislocation was more concentrated and had the most FCC content in the crystal structure.

As shown in Figure 10, many Shockley dislocations formed in the early stage of stretching as the strain increased, which was the leading cause of the stacking faults slipping. Meanwhile, many Frank dislocation and stacking faults formed as the strain increased. For nanopores with different depths for the same strain, when the depth of the nanopores gradually increased from 20 to 30 nm, the Shockley dislocation of the FCC increased. Moreover, the stacking faults of the HCP and the Frank dislocation did not gradually increase. The above analysis indicated that the Shockley dislocation gradually increased as the nanopore depths of the CoCrFeMnNi high-entropy alloy increased. When the depth of nanopores exceeded 10 nm and increased gradually, the Shockley dislocation increased gradually. However, the HCP stacking layer dislocation and Frank stacking faults increased with the depth [30], which corresponded precisely to the analysis of the stresses at the different depths as shown in Figure 6 and the analysis of the stacking faults slipping with the increase in the Shockley dislocation shown in Figure 10.

Figure 11a shows a waterfall chart in which the depth gradually increases from 20 nm to 30 nm, and Shockley dislocation density increases significantly under different strains, which indicates that the Burgers vector 1/6 < 112 > of dislocations was the leading cause of crystal distortion. In addition, when the depth of the nanopores was 10 nm, the density of the dislocations was more than the density of dislocations of the other nanopore depths under different strains. However, combined with the analysis of the tensile stress curve of Figure 6, it shows that the stress at a depth of 10 was higher than those at depths of 20, 30, and 40 nm under different strains, which was because the CoCrFeMnNi high-entropy alloy nanopores at a depth of 10 nm had fewer crystal defects than the deeper nanopores. Otherwise, the volume of the nanopores represents the missing part of the crystal structures. The more profound the nanopores’ depth is, the larger the nanopores’ volume is. Accordingly, the dislocation contents decrease, and the corresponding dislocation densities decrease. A noticeable dislocation junction is observed in Figure 11b–d. Dislocation junctions impede the movement between dislocations and nullify the effects of certain crystal defects [18]. Although the dislocation densities gradually partly increase during the stretching process, the overall dislocation densities of the crystal structures still decrease due to the larger size of the missing part in crystal structures, which is why the stress–strain curves for depths 20, 30, and 40 nm are close together. The analysis of the tensile stress–strain curves at different depths is shown in Figure 6.

### 3.3. Analysis of the Stress Evolution Mechanism of High-Entropy Alloys Containing Nanopores of Different Dimensions and Widths

When the depth of the nanopores was 20 nm, and the strain rate was 05 ps^−1^, the nanopore widths of 10, 20, 30, and 40 nm in the CoCrFeMnNi high-entropy alloy were stretched. The stress–strain curves along the tensile direction were obtained, as shown in Figure 12.

As shown in Figure 12, when the widths of the nanopores were 10, 20, 30, and 40 Å, the corresponding peak stresses in the stress–strain curve were 6.71, 4.33, 2.96, and 2.53 GPa, respectively. The peak stress was gradually increased with the increasing width of the nanopores, and the differences in the peak stress were apparent. As shown in Figure 12, when the strain was less than 0.05, the stress and strain showed a linear correlation that showed apparent elastic behavior. Meanwhile, as the width of the nanopores increased, the elastic modulus decreased. The mechanical properties of the CoCrFeMnNi high-entropy alloy were more sensitive to the nanopore width.

To further study the mechanical deformation regularity of different widths of the nanopores applied to a drug-eluting stent, we analyzed the stress–strain curve using strains of 0.05, 0.1, 0.1, 0.15, and 0.2. The average principal stress cloud map in Figure 13 shows the stress before and after stretching. 

As shown by the average principal stress cloud map in Figure 13, under the same strain, the average principal stress decreased significantly with an increase in the nanopore width, and the average stress was mainly concentrated along the tensile edge and nanopore edges. As the stretching continued, the average principal stress was more focused along the stretching edge of the unit cell system. Compared with the CoCrFeMnNi high-entropy alloy nanopore widths of 20, 30, and 40 nm, the structure with a width size of 10 nm was complete. At the same time, when comparing the width sizes of 20 nm nanopores and 30 nm nanopores, the maximum average principal stresses of the 10 nm nanopores under the same strain were approximately 1.5 times and three times greater, respectively, than those with nanopore widths of 20 and 30 nm.

The average stress gradually increased when the strain increased from 0.05 to 0.15 for the exact widths of the nanopores. However, when the strain was 0.2, the maximum average principal stress decreased by 2.84, 2.50, 2.16, and 1.54 GPa for nanopore widths of 10 to 40 mm, respectively. At the same time, the crystal structure was broken.

#### Influence of Different Nanopore Widths on the Dislocation Evolution of the High-Entropy Alloy

To analyze the influence of the structure and the mechanism of the dislocation evolution alloy with the different widths of nanopores on the mechanical properties, a phase transition analysis of the alloy with different nanopore widths was performed under the tensile strain rate of 0.05 ps^−1^. The structural percentages of different nanopore widths in the CoCrFeMnNi high-entropy alloy crystals with strains of 0.05, 0.1, 0.15, and 0.2 are shown in Figure 14.

Figure 14 shows the crystal structure of the nanoporous CoCrFeMnNi high-entropy alloy, including FCC, HCP, and other crystals. The percentage of the main component FCC was close to 80%. The different crystals were approximately 20%, and the lowest was HCP, which was less than 1%. Through the horizontal comparative analysis shown in Figure 14, with the strain increasing from 0.05 to 0.2, the percentage of FCC structures gradually decreased, and other crystals gradually increased. However, the change in the percentage range was small and did not exceed 1% overall. For strains of 0.05 and 0.2, the percentages of the crystal structures of FCC, other crystals, and HCP are shown in Table 5. As shown, the FCC transformed into other crystals and HCP, but the transition of the crystal structure was not considerable.

The crystal structure of nanopores in the CoCrFeMnNi high-entropy alloy at different depths under different strains was further analyzed, as shown in the RDF waterfall diagram in Figure 15.

The waterfall diagram in Figure 15a–d shows that the characteristic peaks were in the long and short ranges. Furthermore, the long and short ranges were orderly, but the length of the peaks differed. It was a polycrystalline structure. For nanopores of the same width, the types of characteristic peaks increased during the tensile process along with the strain. For different nanopore widths, an increase in nanopore width led to changes in the peak location and peak width of the same atomic RDF for both the short and long ranges. However, the atomic species corresponding to the short-range and long-range peaks did not change.

Upon analyzing the FCC crystal dislocations shown in Figure 16, under the same tensile strain, we found that when the nanopore widths were 20, 30, and 40 nm, the dislocations did not increase considerably. At the same time, when the strain was gradually increased from 0.005 to 0.2, the number of dislocations did not increase notably for the nanopores of the same width size. Otherwise, because the small size of the nanopores of the 10 nm crystal structure had fewer crystal structure defects relative to the 20, 30, and 40 nm nanopores, the Burgers vector 1/6 < 112 > of dislocations primarily occurred in the 10 nm nanopores. In addition, as shown in Figure 16, when nanopore widths were 20, 30, and 40 nm, the stacking faults on the crystal structure increased considerably under the same strain. From the qualitative analysis, we found that because the stacking faults increased with an increase in nanopore width and because the number of dislocations did not change significantly, there were significant differences in peak stress between different widths of the nanopores in the CoCrFeMnNi high-entropy alloy. 

To further analyze the dislocation density in the FCC, we prepared the dislocation density distribution curve for different strains at different nanopore widths, as shown in Figure 17.

In Figure 17a, the waterfall chart shows that the dislocation densities of all species in the FCC crystal structure were relatively lower. As the strain gradually increased, the dislocation densities of nanopores with widths 20, 30, and 40 nm gradually decreased and became close. Meanwhile, the highest dislocation density was 1/6 <112> (Shockley), and the maximum dislocation density was only 0.000438762 A^−2^. This indicates that the dislocation density was not the main factor that affected the mechanical properties of the nanopores of different widths in the CoCrFeMnNi high-entropy alloy. In Figure 17b–d, The occurrence of extrinsic stacking faults (ESFs) and intrinsic stacking defaults (ISFs), as well as the HCP phase, is evident in Figure 10b–d. Combined with the analysis of Figure 16, as is shown in Figure 17b–d, the extrinsic stacking faults, intrinsic stacking faults, and the HCP phase gradually appeared and increased at the edge of the crystal structure along the direction of tensile fracture with increased strain. Meanwhile, the number of intrinsic and extrinsic stacking faults gradually increased with increased strain. In addition, a small number of intrinsic stacking faults converted to extrinsic stacking faults, which produced more lattice defects due to the increase in the stacking faults of the surface.

The above analysis indicated that the dislocation density was low, and the dislocation change was minor. Therefore, for the CoCrFeMnNi high-entropy alloy with different widths, the extrinsic and intrinsic stacking faults increased considerably during the process of stretching, resulting in a large number of face defects appearing in the nanopores in the CoCrFeMnNi high-entropy alloy. Otherwise, as shown in Figure 16, there was no obvious deformation in the nanopores during the stretching process, and the tensile fracture occurred at the crystal structure on both sides of the nanopores in the CoCrFeMnNi high-entropy alloy and along the edge of the tensile direction. When the strain was 0.2, and the width size of the nanopores gradually increased from 10 to 40 nm, Figure 16 shows that the fracture displacement along the tensile direction at the edge of the crystal structure increased considerably. An increase in the width size of the nanopores could lead to more significant structural defects in the CoCrFeMnNi high-entropy alloy crystal structure. The increase in the structural defects is likely to reduce the mechanical properties of the nanopores in the CoCrFeMnNi high-entropy alloy.

## 4. Conclusions

According to the above analysis, with an increase in the depth of the nanopores, the percentage of FCC crystals decreased and converted to HCP, and other crystals, especially FCC crystals, were the main components. The Burgers vector 1/6 < 112 > of dislocations in the FCC crystal structure was the main component with increasing strain. Meanwhile, because the nanopores in the CoCrFeMnNi high-entropy alloy had a large dislocation density, the dislocations hindered the movement of dislocations with increased strain in the stretching process. When the nanopore’s depth changes, the change in the alloy’s mechanical properties is not apparent. The change in the nanopore width had a more significant effect than nanopore depth on the mechanical properties of the CoCrFeMnNi high-entropy alloy. The face defect of the nanoporous CoCrFeMnNi high-entropy alloy crystal structure became larger with an increase in nanopore width. Moreover, the number of intrinsic stacking faults and extrinsic stacking faults gradually increased with increased strain. In addition, a small number of intrinsic stacking faults converted to extrinsic stacking faults, which produced more lattice defects due to the increase in stacking faults of the face. With an increased nanopore width and tensile strain, the percentage of the crystal structure and the types of dislocations did not change. Meanwhile, due to the increase in the nanopore width, the crystal structural defects of the nanopores in the CoCrFeMnNi high-entropy alloy became more extensive, which is much larger than the reduction in the mechanical properties of the nanopores in the alloy because of the stacking fault. According to the above analysis, the nanopore depth of the CoCrFeMnNi high-entropy alloy should be optimized, and the nanopore width should be minimized to maintain the mechanical properties and maximum drug loading limit of the nanopores in the CoCrFeMnNi high-entropy alloy in drug-eluting stents. The study of multiple nanopores on the mechanical properties of CoCrFeMnNi high-entropy alloy, used as a drug-eluting stent, helps guide the processing methods and forms of drug-loaded vascular stents. This research helps determine the relationship between drug-loading capacity and the mechanical properties of drug-loaded vascular stents. However, the scaffold’s mechanical properties are measured, assuming the nanopores are large and numerous enough to have better drug-loading capacity without considering the release of drug-loaded particles inside the nanopores. 

## Figures and Tables

**Figure 1 materials-17-03314-f001:**
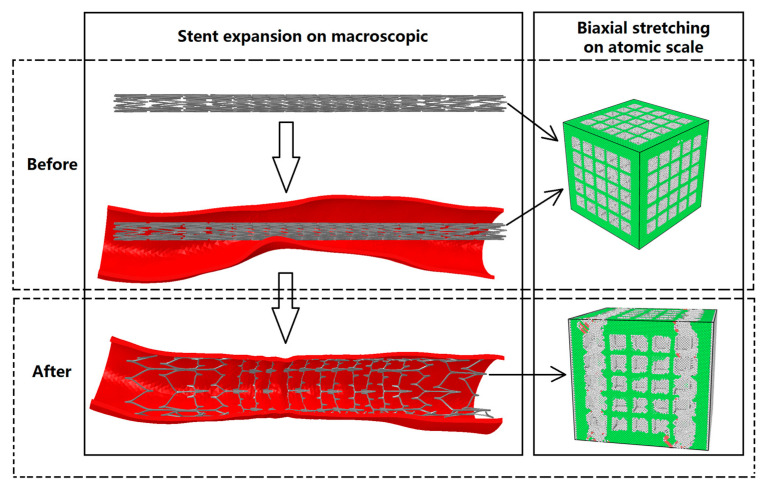
The diagram of stent expansion with macroscopic and biaxial stretching on the atomic scale.

**Figure 2 materials-17-03314-f002:**
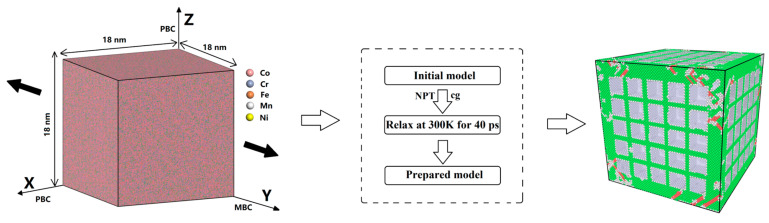
The prepared model includes nanopores with nanopore widths of 20 nm.

**Figure 3 materials-17-03314-f003:**
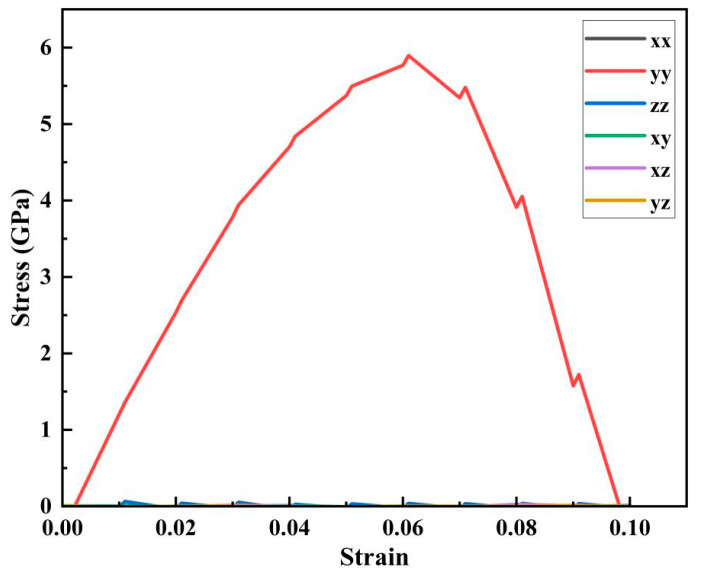
Hexaxial stress–strain curve of the CoCrFeMnNi high-entropy alloy with nanopore widths of 20 nm.

**Figure 4 materials-17-03314-f004:**
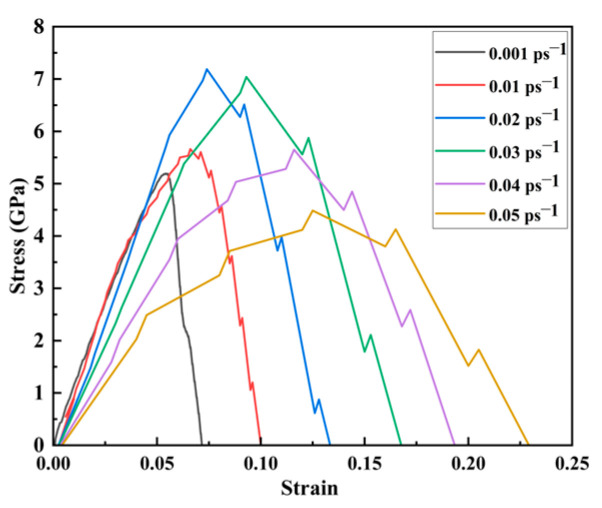
Stress–strain curves under different strain rates with nanopore widths of 20 nm.

**Figure 5 materials-17-03314-f005:**
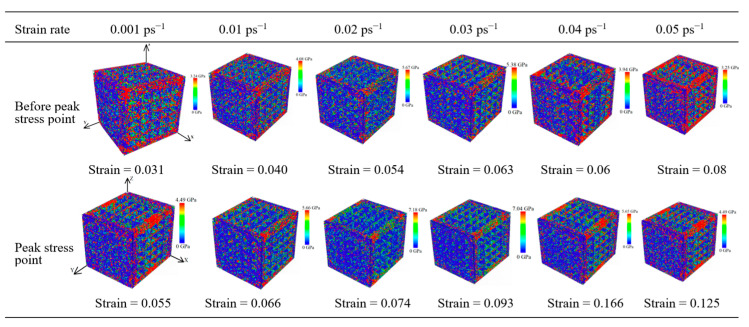
Average principal cloud map in different strain rates.

**Figure 6 materials-17-03314-f006:**
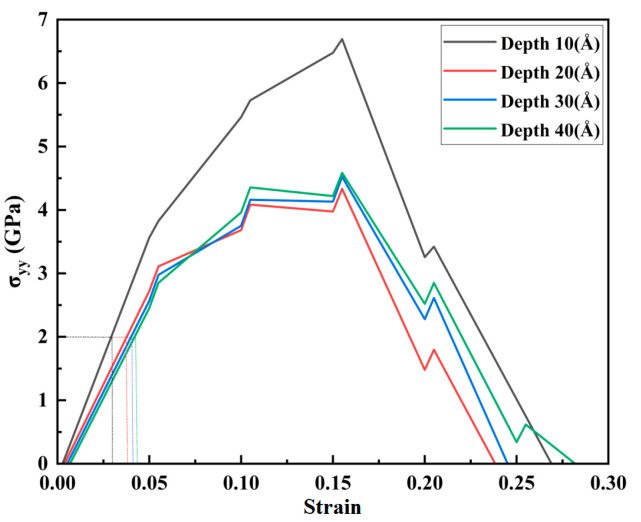
Tensile stress–strain curves at different depths.

**Figure 7 materials-17-03314-f007:**
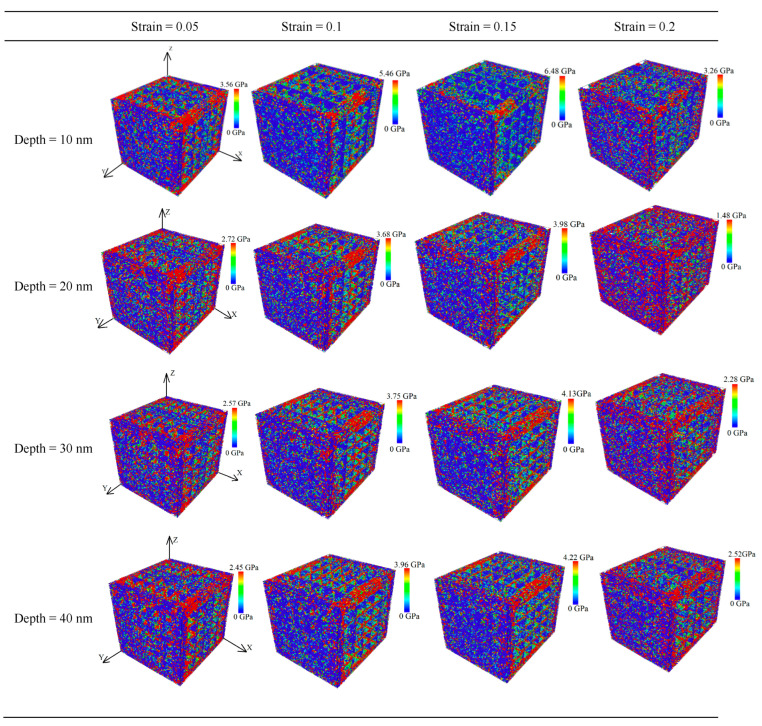
Average principal stress cloud chart of different depths for nanoporous CoCrFeMnNi high-entropy alloy.

**Figure 8 materials-17-03314-f008:**
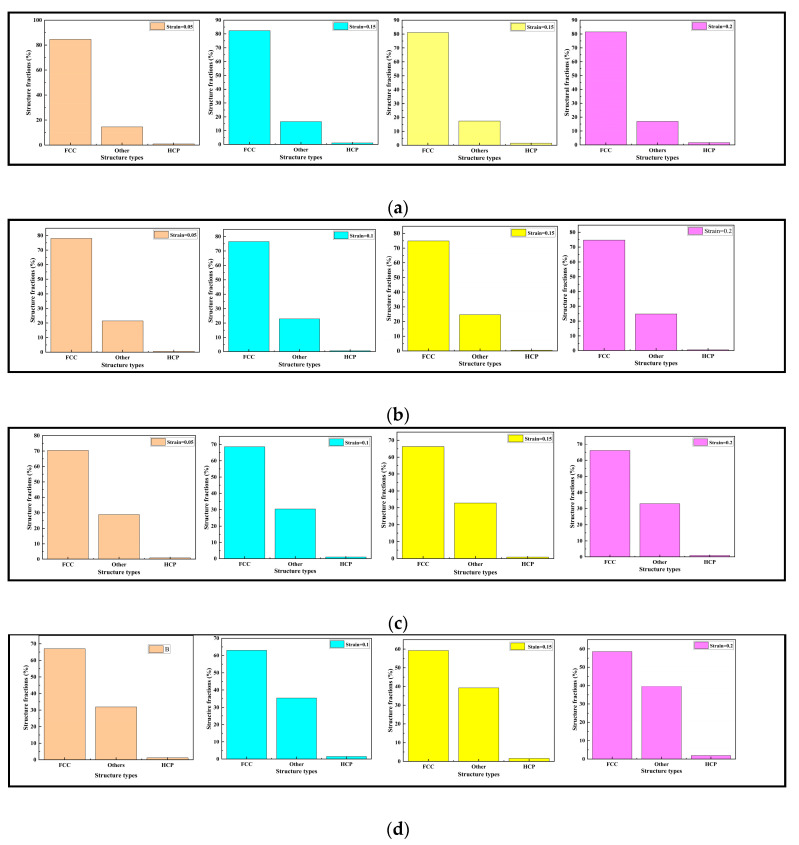
Crystal structure percentages of the nanopores in the CoCrFeMnNi high-entropy alloy with different strains.

**Figure 9 materials-17-03314-f009:**
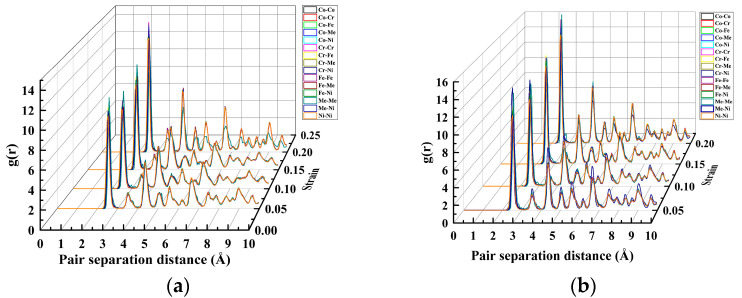
RDF waterfall diagram under different strains and at various depths. (**a**) Depth at 10 nm, (**b**) depth at 20 nm, (**c**) depth at 30 nm, and (**d**) depth at 40 nm.

**Figure 10 materials-17-03314-f010:**
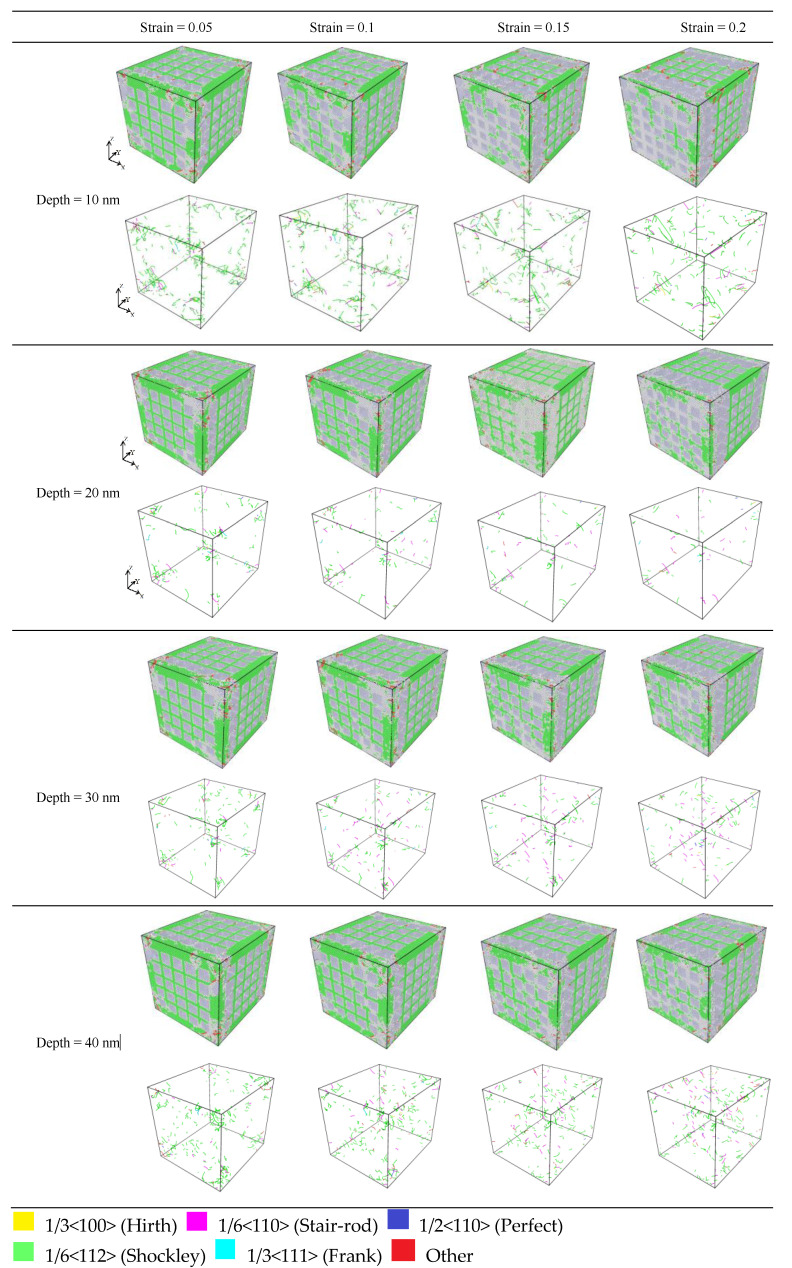
Dislocation of pores of different strains at different depths for the nanoporous CoCrFeMnNi high-entropy alloy.

**Figure 11 materials-17-03314-f011:**
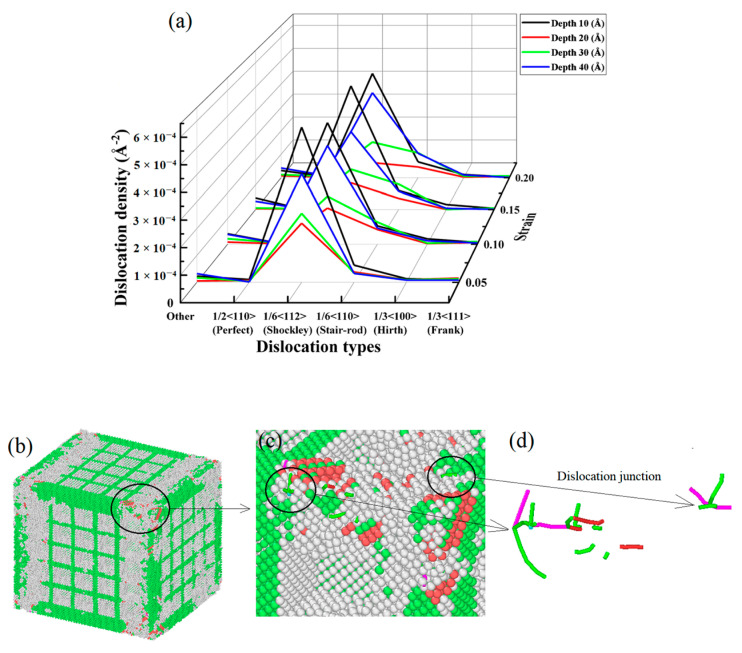
Distribution of dislocation densities for different strains under different nanopore depth sizes.

**Figure 12 materials-17-03314-f012:**
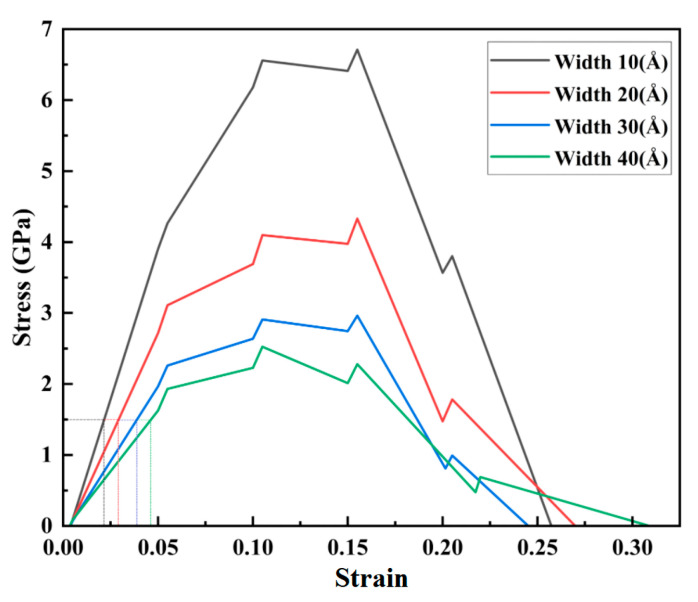
Tensile stress–strain curves at different depths.

**Figure 13 materials-17-03314-f013:**
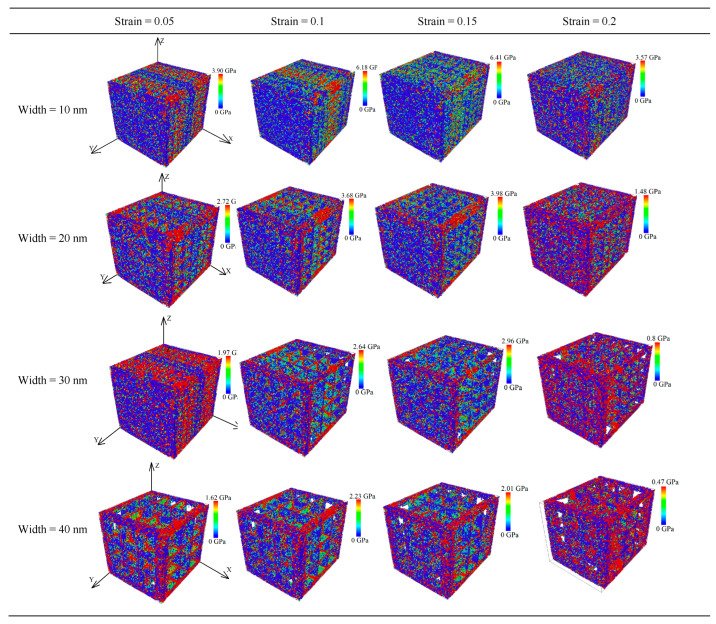
Average principal stress cloud chart of different widths for nanoporous CoCrFeMnNi high-entropy alloy.

**Figure 14 materials-17-03314-f014:**
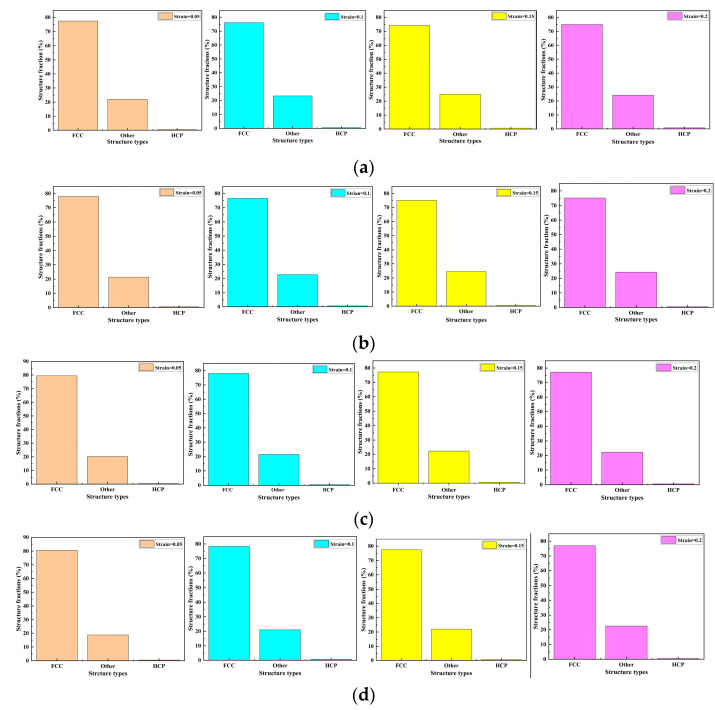
Crystal structure percentages of the nanopores in the CoCrFeMnNi high-entropy alloy with different strains. (**a**) Width size = 10 nm. (**b**) Width size = 20 nm. (**c**) Width size = 30 nm. (**d**) Width size = 40 nm.

**Figure 15 materials-17-03314-f015:**
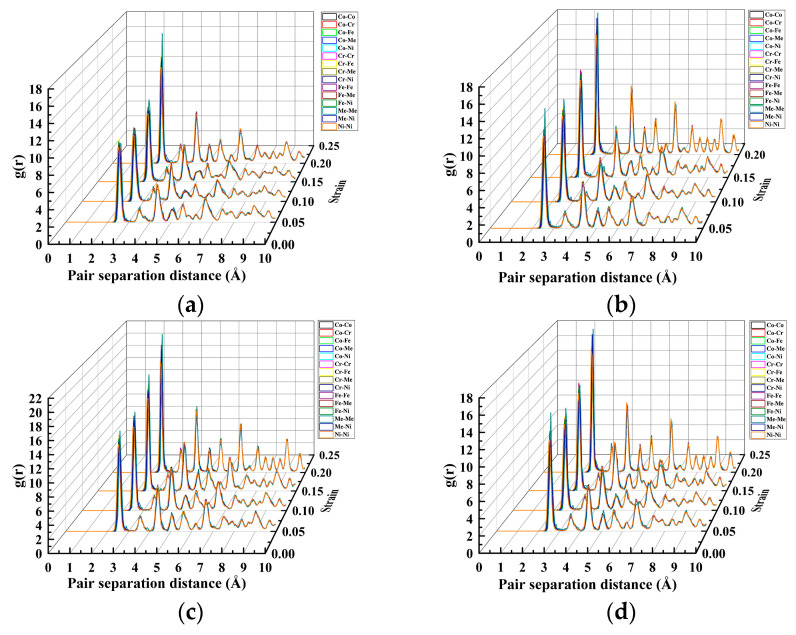
RDF waterfall diagram under different strains and widths. (**a**) Width at 10 nm. (**b**) Width at 20 nm. (**c**) Width at 30 nm. (**d**) Width at 40 nm.

**Figure 16 materials-17-03314-f016:**
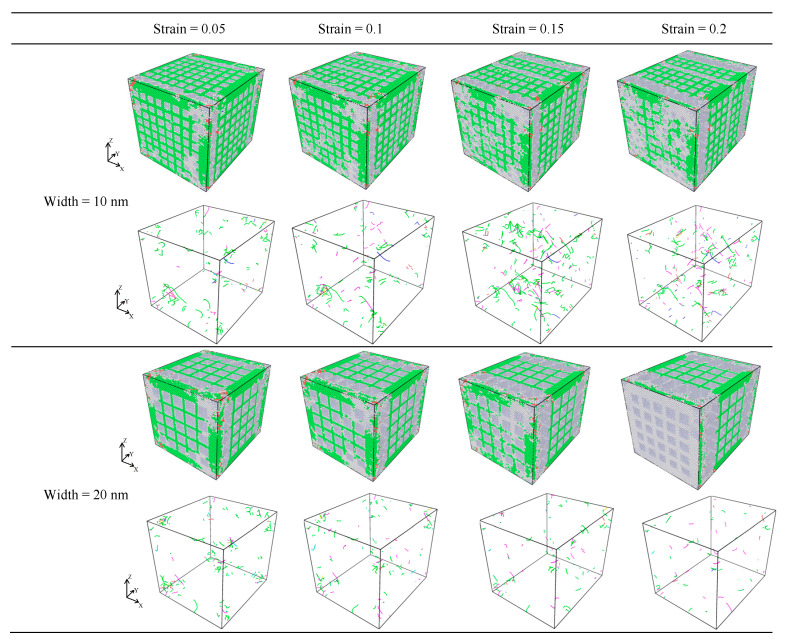
Dislocation of nanopores under different strains at different widths for nanoporous CoCrFeMnNi high-entropy alloy.

**Figure 17 materials-17-03314-f017:**
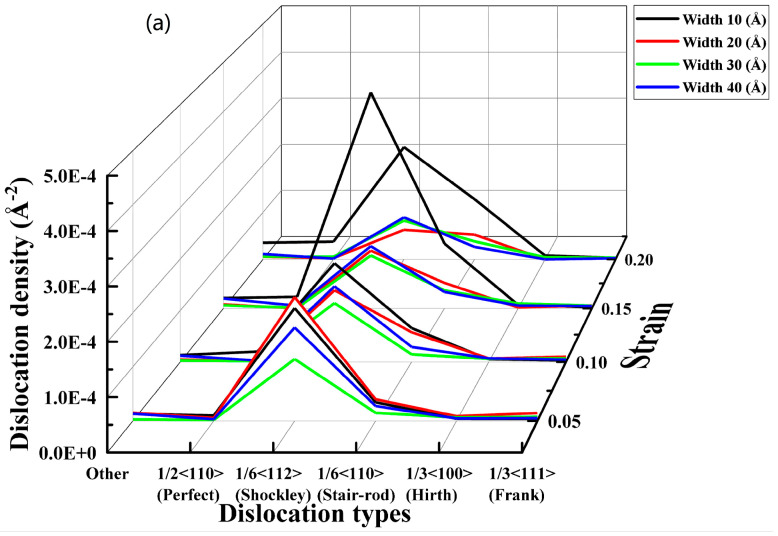
Distribution of dislocation density for different strains at different nanopore widths.

**Table 1 materials-17-03314-t001:** Parameters of the potential function of the CoCrFeMnNi high-entropy alloy [22].

Element 1	Element 2	$\sigma \;\left[Å\right]$	$\varepsilon \;\left[Å\right]$	$r_{cut}\;\left[Å\right]$
Co	Co	2.271	0.227	
Cr	Cr	2.315	0.222	
Fe	Fe	2.301	0.232	6.60
Mn	Mn	2.505	0.122	
Ni	Ni	2.264	0.230	
Co	Cr	2.293	0.225	
Co	Fe	2.286	0.230	6.60
Co	Mn	2.388	0.166	
Co	Ni	2.268	0.229	
Cr	Fe	2.308	0.227	6.60
Cr	Mn	2.410	0.164	
Cr	Ni	2.289	0.226	
Fe	Mn	2.403	0.168	6.60
Fe	Ni	2.283	0.231	
Mn	Ni	2.384	0.167	6.60

**Table 2 materials-17-03314-t002:** Elemental composition of the CoCrFeMnNi high-entropy alloy [20].

Element	Cr	Fe	Mn	Ni	Co
Content (%)	0.2	0.03	0.015	0.1	Balance

**Table 3 materials-17-03314-t003:** Nanopores of different sizes under different tensile rates.

Box Dimension (Å)	Pore Width (nm)	Pore Depth (nm)	Tensile Rate
	40	40	0.0001/Ps
180 × 180 × 180	30	30	0.001/Ps
	20	20	0.002/Ps
	10	10	0.003/Ps
	-	-	0.004/Ps
	-	-	0.005/Ps

**Table 4 materials-17-03314-t004:** Classification analysis of thrombosis under different conditions.

Analysis Classification	Number	Pore Width (nm)	Pore Depth (nm)	Strain Rate (ps^−1^)
Different tensile rate	①	20	20	0.0001
	②	20	20	0.001
	③	20	20	0.002
	④	20	20	0.003
	⑤	20	20	0.004
	⑥	20	20	0.005
Different pore width	⑦	10	10	0.005
	⑧	20	10	0.005
	⑨	30	10	0.005
	⑩	40	10	0.005
Different pore depth	⑪	20	10	0.005
	⑫	20	20	0.005
	⑬	20	30	0.005
	⑭	20	40	0.005

**Table 5 materials-17-03314-t005:** Percentages of FCC, HCP, and other crystals in the CoCrFeMnNi high-entropy alloy.

Width	Structure	Fraction (%)		Variable (%)
		Strain (0.05)	Strain (0.2)	
10 nm	FCC	77.4	75	−2.4
	Other	22.1	24.2	+2.1
	HCP	0.5	0.8	+0.3
30 nm	FCC	78	75.2	−2.8
	Other	21.4	24.3	+2.9
	HCP	0.6	0.5	−0.1
40 nm	FCC	79.4	77.2	−2.2
	Other	20.2	22.2	+2.0
	HCP	0.4	0.6	+0.2
50 nm	FCC	80.62	76.9	−3.72
	Other	18.83	22.6	+3.77
	HCP	0.544	0.5	−0.044

+ = increase. − = decrease.

## Data Availability

The data presented in this study are available on request from the corresponding author, upon reasonable request.

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
