# Peer review of "Study on Mechanical Properties of Nanopores in CoCrFeMnNi High-Entropy Alloy Used as Drug-Eluting Stent"

_materials, 2024, doi:10.3390/ma17133314_

Round 1

Reviewer 1 Report

Comments and Suggestions for Authors

The manuscript if of interest for the present journals, Materials, as it reports on vascular stents based on high-entropy alloy ( CoCrFeMnNi) that can potentially release specific drugs. Such stents, thus, can inhibit inflammatory processes around itself, thus inhibiting re-narrowing of the vessel into which they were implanted.

However, the manuscript, as it is, cannot be accepted and published and needs revision. For example, it is not very clear to what extent the reported results can be applied to stents made of other alloys. Nevertheless, after proper revision, the manuscript can be reconsidered. Please see the comments and suggestions below:

1\     Why did the authors decide to model the alloy composition mentioned in the title? On what basis was this particular alloy composition was used (Table 2)? Also, a related question: to what extent the reported results can be applied to other alloys, more specifically, to stents made from other alloys?

2\      What volume of a drug can be captured by (or loaded into) the developed alloy (as a porous stent) and is this volume significant for the solution of the anti-inflammatory task?

3\      Young's modulus of the studied materials is expected to be about 100-150 GPa. How  large value this is  and how it correlates with experimental work(s) reported so far by any other group?

4\    What are the reasons for the stress level rises at material failure exhibited in Figures 3-5?

5\     When testing specimens in the Y-Y direction (along Y-Y direction), the "no stresses" assumption seems quite disputable. Are there any explanations to support this assumption?

6\    In general, there is a lack of comparison with experimental specimens. It it not clear to what extent the obtained results related to real experimental results obtained for any similar (or related) materials that also release drugs upon placement in blood vessels.

7\     It is not very clear what shape the pores on the stent surface have. The authors should probably specify it a bit clearer: for instance, square-shaped pores with a certain depth? It is recommended that the authors specify this.

8\   Introduction: it is not mentioned in the Introduction what drugs are usually used on this type of stents (are used or could be used). Chemically –wise and in what form (as particles, as nanoparticles ?). And how, in principle, such drugs can be loaded into such stents, i.e., inside their pores. Just a few words, 1-2 general sentence in this regard would be enough. The authors should remember that not all potential readers are from the same area, so they may need such brief clarification.

9\ Technically, Tables 5-8 and 10 are not tables but figures. It is recommended that the authors convert them to figures. It is quite confusing to see figures treated as tables.

10\ Minor comment: Reference list must be properly organized, according to MDPI requirements/

Comments on the Quality of English Language

The authors should read the entire manuscript and polish its language.

Author Response

Question 1: Why did the authors decide to model the alloy composition mentioned in the title? On what basis was this particular alloy composition was used (Table 2)? Also, a related question: to what extent the reported results can be applied to other alloys, more specifically, to stents made from other alloys?

Answer: We are grateful for your valuable suggestion. Your every question help us to improve our article.  Here are our response. The material high-entropy alloy composition is used by material L605, cited in the literature [18]. The application note is marked in Table 2. Because of its excellent mechanical properties, high-entropy Alloy L605 is often used as a coronary vascular stent material. The cited article in our article “[18] Thithuha Phan, John E. Jones, Meng Chen, Doug K. Bowles. 2022. A biocompatibility study of plasma Nanocatings onto cobalt chromium L605 Alloy for cardiovascular stent applications. Materials. 15: 5968”. Compared to stainless-steel stents, high-entropy alloy L605 stents have been widely used because of their excellent mechanical and radiopaque properties and good biocompatibility. Thin-strut stent design reduced restenosis and thrombosis events after stent implantation. Despite containing thinner struts compared to stainless-steel stents, CoCr L605 stents have minimal elastic recoil while maintaining X-ray visibility during and after percutaneous coronary intervention. The CoCrFeMnNi high-entropy alloy is compatible and supports smooth muscle cells (SMCs) triggered by vascular injury. 

Question 2: What volume of a drug can be captured by (or loaded into) the developed alloy (as a porous stent), and is this volume significant for the solution of the anti-inflammatory task?

Answer: The article “Ho-Jae Kanga, Sung-Joon Parkb, Ji-Beom Yooc, Deug Joong Kim. 2007. Controlled Drug release using Nanoporous Anodic Aluminum Oxide. Solid State Phenomena. 709:121-123.” has explained the nanoscale volume of a drug. First of all, it is made into an aqueous solution of the drug with a certain proportion and then and then treated with an ultrasonic for the loading of the drug. In this way, the volume of the drug carried in the nanopore is based on the size of the nanopore on the stent. This article has explained the effect of the nanopore diameter and depth on the drug loading capacity, especially since the large pore diameter is more favorable for drug loading. Many articles like this only explained that the large pore diameter is more favorable for drug loading. However, these articles focused on the more effective drug loading capacity with large pore diameter and often ignored the stent's mechanical properties. That’s why I’m writing this article from a molecule dynamic perspective.

Question 3: Young's modulus of the studied materials is expected to be about 100-150 GPa. How large value this is  and how it correlates with experimental work(s) reported so far by any other group?

Answer: Compared the article “Paul Lohmuller, Laurent Peltier, Alain Hazotte. 2018. Variations of the Elastic Properties of the CoCrFeMnNi High Entropy Alloy Deformed by Groove Cold Rolling. Materials. 11(8): 1337”, the Young's modulus is about 175 GPa during grooved cold rolling. Due to the limitation of molecular dynamics simulations and the influence of the volume fraction of each element in the simulation process, our simulation results are acceptable.

Question 4: What are the reasons for the stress level rises at material failure exhibited in Figures 3-5?

Answer: From Figures 3-5, the CoCrFeMnNi high-entropy alloy failure mode is a typical ductile fracture mode. From figures 3-5, we can see the different stress states at other strains. From Tables 6-8, we can see the different stress contours at different stress strains. When the strain is 0.15, the fracture strength is the maximum. When the strain is from 0.15 to 0.2, the ends of the crystal model gradually break under the action of stretching. The mode of failure is the gradual breakage under tensile action.

Question 5: When testing specimens in the Y-Y direction (along Y-Y direction), the "no stresses" assumption seems quite disputable. Are there any explanations to support this assumption?

Answer: Because the tensile direction is Y-Y, the tensile hexaxial stress-strain curve in Figure 3 displays that the stress in the other directions is almost 0 GPa. We only focused on the braid of the tensile direction. The article “Yuming Qi, Heming Xu, Tengwu He. 2021. Atomistic simulation of deformation behaviors polycrystalline CoCrFeMnNi high-entropy alloy under uniaxial loading. International Journal of Refractory Metals and Hard Materials. 95(2021):105415” also only consider the stress of the tensile direction. This article explains to support this.

Question 6: In general, there is a lack of comparison with experimental specimens. It it not clear to what extent the obtained results related to real experimental results obtained for any similar (or related) materials that also release drugs upon placement in blood vessels.  

Answer: We answer your good question, “Young's modulus of the studied materials is expected to be about 100-150 GPa. How  large value this is  and how it correlates with experimental work(s) reported so far by any other group?” that has helped us prove that the result of the simulation is convincing. Due to the volume fraction of the elements during the simulation as well as the relaxation method, our Young’s modulus is close to the experiment level in the article “Paul Lohmuller, Laurent Peltier, Alain Hazotte. 2018. Variations of the Elastic Properties of the CoCrFeMnNi High Entropy Alloy Deformed by Groove Cold Rolling. Materials. 11(8): 1337”.

Question 7: It is not very clear what shape the pores on the stent surface have. The authors should probably specify it a bit clearer: for instance, square-shaped pores with a certain depth? It is recommended that the authors specify this.

Answer: We are grateful for your valuable suggestion. Your advice is very professional, and we have made modifications.

Question 8: Introduction: it is not mentioned in the introduction what drugs are usually used on this type of stents (are used or could be used). Chemically–wise and in what form (as particles, as nanoparticles ?). And how, in principle, such drugs can be loaded into such stents, i.e., inside their pores. Just a few words, 1-2 general sentence in this regard would be enough. The authors should remember that not all potential readers are from the same area, so they may need such brief clarification.

Answer: We are very grateful for your valuable suggestion. The corresponding sentences have been added to the introduction.

Question 9: Technically, Tables 5-8 and 10 are not tables but figures. It is recommended that the authors convert them to figures. It is quite confusing to see figures treated as tables.

Answer: We are very grateful for your valuable suggestion. We have corrected these.

Question 10: Minor comment: Reference list must be properly organized, according to MDPI requirements

Answer: We are very grateful for your valuable suggestion, and we have correctly organized the reference list for MDPI requirements.

Reviewer 2 Report

Comments and Suggestions for Authors

Study on Mechanical Properties of Nanopores in CoCrFeMnNi  High-entropy Alloy Used as Drug-eluting Stent

 materials-2979206

 Manuscript could be accepted after minor corrections:

 1.. The novelty of a study should be clearly highlighted, often in the introduction or discussion sections, to underscore its unique contributions to the field.

2. Please strengthen the literature review and expand on how this work builds upon or differs from existing research, emphasizing the unique contribution.

3. A more comprehensive discussion on the limitations of the current study and potential areas for future research would strengthen the manuscript

Comments on the Quality of English Language

Minor improvement is necessary

Author Response

Question 1: the novelty of a study should be clearly highlighted, often in the introduction or discussion sections, to underscore its unique contributions to the field.

Answer: We are very grateful for your valuable suggestion, and the study's novelty has been highlighted in the introduction. To better understand the effect of multiple surface nanopores on the mechanical properties of vascular drug-loaded stents, this paper analyzes the mechanics and microstructure evolution of nanoporous drug-loaded scaffolds in the process of expansion from the perspective of molecular dynamics using high-entropy alloys with square group pores of different depths and widths. The high-entropy alloy CoCrFeMnNi was selected as the research material because material L605 uses the high-entropy alloy composition. The high-entropy alloy L605 is compatible and supports smooth muscle cells triggered by vascular injury [18]. Compared to stainless-steel stents, high-entropy alloy L605 stents have been widely used because of their excellent mechanical and radiopaque properties and good biocompatibility [18].

Question 2: Please strengthen the literature review and expand on how this work builds upon or differs from existing research, emphasizing the unique contribution.

Answer: to strengthen the literature review and expand on how this work builds upon or differs from existing research, we add the literature that analyzes the effect of voids and HCP-Phase inclusion on the deformation of single-crystal CoCrFeMnNi high-entropy alloy. This leads to our study of the mechanical properties of multiple pores [18-19].

[18] Yuming Qi; Xiuhua Chen; Miaolin Feng. Molecular dynamics-based analysis of the effect of voids and HCP-Phase inclusion on deformation of single-crystal CoCrFeMnNi high-entropy alloy. Materials Science and Engineering: A, 2020, 791, 139444, doi:10.1016/j.msea.2020.139444

[19] Yuhang Qiu; Yuming Qi; Huayong Zheng; etc. Atomistic simulation of nanoindentation response of dual-phase nanocrystalline CoCrFeMnNi high-entropy alloy. Journal of Applied Physics, 2021, 130, 125102, doi:10.1063/5.0057591

Question 3: a more comprehensive discussion on the limitations of the current study and potential areas for future research would strengthen the manuscript.

Answer: We are very grateful for your valuable suggestion and for a more comprehensive discussion on the limitations of the current study and potential areas for future research in the conclusion. This research helps determine the relationship between drug-loading capacity and the mechanical properties of drug-loaded vascular stents. However, the scaffold's mechanical properties are measured, assuming the nanopores are large and numerous enough to have better drug-loading capacity without considering the release of drug-loaded particles inside the nanopores.

Reviewer 3 Report

Comments and Suggestions for Authors

The paper titled “Study on Mechanical Properties of Nanopores in CoCrFeMnNi High-entropy Alloy Used as Drug-eluting Stent” offers valuable insights into optimizing the mechanical properties and drug-carrying capacity of high-entropy alloy stents by means of a high-entropy alloy system with different nanopore distributions created using molecular dynamics. The manuscript's scope and structure are well-defined and comprehensible. However, some minor revisions are recommended prior to publication:

Introduction:

Include information about the type and size of drug particles in the state of the art.

Line 71: Define MD (Molecular Dynamics) as it appears for the first time.

Modeling and simulation methods:

Line 111: Add the caption for Table 1 after line 117.

Table 2: Ensure font size consistency with the rest of the article. How was this chemical composition obtained/measured?

Table 3: Ensure font size consistency with the rest of the article. Provide explanations for selecting pore depth and width.

Lines 129-132: Rewrite to avoid repetition of concepts.

Results and discussion:

Equation 5: Check font size consistency.

Table 5: Color scale should be the same for each map, and it must be readable. As a suggestion consider including a unique larger scale for the entire table.

Figures 2, 3 and 4 include nanopores? Please indicate which are pore sizes shown in these examples.

Lines 204-205: You state that “we found that the more significant strain rate was 0.05 ps−1”, why? Provide an explanation and why have you selected this strain rate.

Table 6, Table 8: Color scale should be the same for each map, and it must be readable. As a suggestion consider including a unique larger scale for the entire table.

Line 228: Reformulate the sentence for clarity.

Figure 6 and Figure 10: Increase font size for graph axes information. Define conditions (a) to (d).

Figure 7 and Figure 11: Include RDF = Radial Distribution Function in the figure caption and define conditions (a) to (d) (e.g., Nanopore depth at 10 nm).

Figure 8 and Figure 12: Include (a)-(d) description in figure caption.

Author Response

Question 1: The novelty of a study should be clearly highlighted, often in the introduction or discussion sections, to underscore its unique contributions to the field.

Answer: We are very grateful for your valuable suggestions. Every one of your prudent and valuable suggestions has been revised in the revised manuscript. Thank you very much.

Round 2

Reviewer 1 Report

Comments and Suggestions for Authors

The manuscript has been improved by the authors and now it can be accepted

Comments on the Quality of English Language

The authors are recommended to polish their manuscript, getting rid of small mistakes and typing errors here and there